# Network Pharmacology of Ginseng (Part III): Antitumor Potential of a Fixed Combination of Red Ginseng and Red Sage as Determined by Transcriptomics

**DOI:** 10.3390/ph15111345

**Published:** 2022-10-30

**Authors:** Alexander Panossian, Sara Abdelfatah, Thomas Efferth

**Affiliations:** 1EuroPharma USA Inc., Green Bay, WI 54311, USA; 2Phytomed AB, 58344 Vastervick, Sweden; 3Department of Pharmaceutical Biology, Institute of Pharmaceutical and Biomedical Sciences, Johannes Gutenberg University, 55131 Mainz, Germany

**Keywords:** red ginseng, red sage/danshen, gene expression, IPA pathways, network pharmacology, transcriptomics, cancer

## Abstract

Background: This study aimed to assess the effect of a fixed combination of Red Ginseng and Red Sage (RG–RS) on the gene expression of neuronal cells to evaluate the potential impacts on cellular functions and predict its relevance in the treatment of stress and aging-related diseases and disorders. Methods: Gene expression profiling was conducted by transcriptome-wide mRNA microarray analyses of murine HT22 hippocampal cell culture after treatment with RG–RS preparation. Ingenuity pathway analysis (IPA) was performed with datasets of significantly upregulated or downregulated genes and the expected effects on the physiological and cellular function and the diseases were identified. Results: RG–RS deregulates 1028 genes associated with cancer and 139 with metastasis, suggesting a predicted decrease in tumorigenesis, the proliferation of tumor cells, tumor growth, metastasis, and an increase in apoptosis and autophagy by their effects on the various signaling and metabolic pathways, including the inhibition of Warburg’s aerobic glycolysis, estrogen-mediated S-phase entry signaling, osteoarthritis signaling, and the super-pathway of cholesterol biosynthesis. Conclusion: The results of this study provide evidence of the potential efficacy of the fixed combination of Red Ginseng (*Panax ginseng* C.A. Mey.) and Red Sage/Danshen (*Salvia miltiorrhiza* Bunge) in cancer. Further clinical and experimental studies are required to assess the efficacy and safety of RG–RS in preventing the progression of cancer, osteoarthritis, and other aging-related diseases.

## 1. Introduction

Various stresses/stressors of exogenous origin, such as environmental carcinogens, chemicals, and radiation, and others endogenously derived from the endoplasmic reticulum, such as hypoxia and oxidative stress, disrupt homeostasis and trigger the progression of many diseases, including cancer. With age, the accidence of cancer, cardiovascular, neurodegenerative, and some other diseases dramatically increases due to the systemic progression of aseptic chronic low-grade inflammation defined as inflammageing [1,2,3]. Inflammaging is neither harmful nor beneficial by itself and may be helpful in an adaptive manner or harmful in a maladaptive way, depending on environmental stressors and genetic background [1]. The human organism has developed a highly effective defense regulatory network, the neuroendocrine-immune complex, to adapt to these stresses. Adaptation to stress/stressors could finally lead to either beneficial or unfavorable consequences where the outcome may be aging-related diseases or healthy longevity [1].

Two well-known adaptogenic plants [4,5,6,7,8,9], Red Ginseng (*Panax ginseng* CA Mey) [10] and Red Sage/Danshen (*Salvia miltiorrhiza* Bunge) [11], are used in Traditional Chinese Medicine (TCM) as an effective treatment of diseases and conditions associated with inflammaging and aging-related diseases [4,5,6,7,8,9,10,11,12,13,14,15,16,17]. In Europe, Red Ginseng (RG) is approved as an herbal medicinal product to enhance cognitive functions and physical capacities in weakness, loss of concentration, exhaustion, tiredness, and convalescence [10]. According to ancient records, Red Ginseng enhances longevity with long-term intake [17]. The antitumor, cardioprotective, and neuroprotective effects of both plants were demonstrated in numerous in vivo and in vitro experimental studies [14,15,16,17,18,19,20,21,22,23,24], suggesting their usefulness in the treatment of aging-related disorders and diseases, including dysfunctions of the central nervous system (CNS), particularly those associated with Alzheimer’s disease, Huntington’s disease, Parkinson’s disease, traumatic brain injury, and ischemia. For example, ginsenosides modulate plentiful physiological and intracellular processes in the brain, including attenuation of oxidative stress, neuroinflammation, excitotoxicity, apoptosis, modulation of adaptive signaling pathways, maintenance of mitochondrial stability, and neurotransmitter homeostasis [21,22,23,24]. Regulations of these pathophysiological processes are associated with the beneficial effects of ginsenosides on cognitive function and stress-protective effects on the brain functions associated with neuroinflammation and neurodegenerative diseases [4,5,21,22,23,24].

The results of clinical trials suggested the beneficial effects of Ginseng on stress and cognitive functions [25,26,27,28] of healthy subjects [29,30,31,32,33,34], and patients with mild cognitive impairments and neurological disorders [17,35,36,37,38,39]. Some clinical studies suggest that both plants are promising treatments for aging-related diseases, including neurodegenerative diseases, cardiovascular diseases, diabetes, and cancer [10,11]. That is in line with the traditional use of Red Korean Ginseng as a panacea for an extraordinary number of disease states normalizing body functions and strengthening body systems compromised by a wide variety of environmental assaults and emotional conditions, presumably due to the recovery of vital energy, alleviation of fatigue, tonic and overall protective effect on health [4,5,8,9].

The fixed combination of the extracts of Red Ginseng (RG) and Red Sage (RS), RG–RS, is known to have favorable cardiovascular effects and is adopted in TCM for treating cardiovascular disease [40,41]. RG–RS ameliorated the delayed onset of vascular stiffening resulting in a faster recovery of muscle soreness, and it prevented an increase in blood C-reactive protein and IL-6 induced by acute downhill running exercises of healthy subjects after supplementation for seven weeks in a daily dose corresponding to 1.75 g of each herbal substance (dry herb) [40]. However, at a lower dose (0.75 g of each herb), RG–RS supplementation for 12 weeks does not appear to modulate vascular and inflammatory adaptations to eccentric exercise training in middle-aged and older adults [41].

Several in vitro and in vivo [42,43,44,45] studies of the so-called “optimized” component formula” (OCF), composed of three active constituents of the extract, salvianolic acids, ginsenosides, and ginseng polysaccharides (5, 10, and 5 mg/L, respectively), in Lewis lung cancer cells (LLC) allografted in C57BL/6 mice and A549 xenografted in nude mice experiments suggested the potential efficacy in preventing cancer progression and tumor metastasis [42,43,44,45]. The OCF inhibited lung cancer cell proliferation and induced apoptosis without any cytotoxic effects on normal lung epithelial BEAS-2B cells. Furthermore, OCF inhibited lung cancer cell migration and invasion. OCF significantly promoted phosphatase and tensin homolog (p-PTEN) expression, which was followed by inhibiting the PI3K/AKT signaling pathway [43].

However, the results obtained in experiments with RG, RS, and OCF cannot be mechanistically extrapolated on RG–RS because that is a different substance with a different chemical composition consisting of 17 active constituents from RG and 53 active constituents from RS [46], which can interact synergistically and antagonistically in the target cells of the organism on transcriptomic and proteomic regulation levels of the cellular metabolism.

Therefore, we aimed in this study for the first time to uncover the mechanism of action (MOA) of RG–RS by assessing its effects on gene expression in isolated neuronal cells, followed by network pharmacology using Ingenuity Pathway Analysis (IPA) of significantly upregulated or downregulated genes, to disclose effects on cellular functions and diseases. That is important for understanding and predicting the potential risks and benefits of the therapeutic action of RG–RS.

The choice of murine neuronal HT22 cells as an in vitro model is based on many publications where a neuroprotective effect of Red Ginseng extracts [47,48,49], Ginsenosides Rg5, Rb2, Rg1, Re [50,51,52,53], Compound K [54,55] and tanshinons I and II [56,57] in various experiments with murine hippocampal HT22 cells was demonstrated. Furthermore, Red Ginseng root preparation HRG80TM (RG, used in this study) modulates ionotropic glutamate NMDA, and kainate receptors mediated transmission in ex vivo experiments of the excitability of hippocampal pyramidal cells of rats [58]. Finally, it was effective in preventing and mitigating the stress-induced deterioration of cognitive functions in healthy subjects [29] and elderly patients with mild cognitive disorders, substantially affecting various brain regions depending on the mental load during relaxation and cognitive tasks associated with memory, attention, and mental performance, suggesting an improvement in mood and calming effects associated with GABA-ergic neurotransmission [39].

## 2. Results

### 2.1. Effect of RG–RS on the Gene Expression Profile in the Hippocampal Neuronal Cell Line HT22

Table 1 shows the number of genes deregulated (>20-fold compared to control) by RG–SG, RG, and ginsenosides Rb1, Rg3, Rg5, and Rk1 in the hippocampal neuronal cell line HT22; for details, see Appendix A and Table 1.

The combination RG–RS has a more substantial impact than RG on gene expression in hippocampal neurons, deregulating almost three-fold (1151/394) more genes, Figure 1.

The profiles (signatures) of deregulated genes by RG–RS and RS are quite different; only 24% of the genes (95 of 397) deregulated by RG were deregulated by the combination RG– RS Figure 1 and Appendix A).

These observations also show that the gene expression profile of hippocampal neurons is specific for the combination RG–RS and quite different from RG extract.

### 2.2. Effects of RG–RS on Signaling Canonical Pathways

Figure 2 and Appendix A show the predicted effects (−log *p*-value > 1.3, z-score > 2) of RG–RS on the canonical signaling pathways, including:Inhibition of glycolysis, estrogen-mediated S-phase entry, PCP (planar cell polarity), mevalonate, osteoarthritis pathways, cholesterol, and geranylgeranyl diphosphate biosynthesis super-pathways, andActivation of MIF regulation of innate immunity, regulation of cellular mechanics by calpain protease, hypoxia signaling in the cardiovascular system, and xenobiotic metabolism CAR signaling pathways.

A large part (about 89%) of deregulated genes consist of networks significantly associated with cancer (Table 1, Figure 3 and Appendix A). The IPA analysis shows the predicted (activation score z > [±2], −log *p*-value > 1.3) inhibition of diseases, Figure 4 and Appendix A.

The largest part of deregulated genes (1011 genes, about 88%) consist of networks significantly associated with decreased tumorigenesis (Figure 4 and Appendix A), 139 genes (12%) with reduced metastasis, 81 (7%) with increased autophagy, and 312 (27%) with increased apoptosis (Figure 5 and Appendix A).

## 3. Discussion

This study’s primary aim was to employ transcriptomics of neuronal cells to uncover the mechanisms of action (MOA) and potential pharmacological activities of a fixed combination RG–RS, consisting of two adaptogenic plants traditionally used in aging-related diseases and disorders. The MOA of both plants and many of their active constituents have been described in numerous publications. However, they might be quite different if both plants are combined in one formulation due to various synergistic and antagonistic interactions within the organism. Consequently, pharmacological and toxicological profiles might be considerably diverse in purified constituents. The present study confirms this common rule, as evident from Figure 1 and Table 1; e.g., RG–RS has to impact the expression of 302 genes specifically deregulated by RG. Furthermore, the gene expression profiles activated or inhibited the signaling pathways and predicted that the pharmacological effects are different (see Figure 2, Figure 3 and Figure 4 and Appendix A).

To our best knowledge, only two clinical studies have been conducted with the RG–RS combination [40,41], suggesting its efficacy in the recovery of muscle soreness and preventing an increase in proinflammatory protein in exercising healthy subjects. Several in vitro and in vivo [42,43,44,45] studies conducted with a combination of purified active constituents of the plant extracts, i.e., salvianolic acids, ginsenosides, and ginseng polysaccharides, suggest the potential efficacy in preventing cancer progression and tumor metastasis [42,43,44,45], which are presumably associated with an inhibition of the PI3K/AKT signaling pathway [43]. From the chemical point of view, the substance RG–RS studied in our experiments differs from the substance used in any other study of RG, RS, and their constituents, except for two clinical studies in athletes. RG consists of nearly 300 compounds (more than 200 have been isolated from Red Ginseng and more than 100 from Red Sage), while OCF [42,43,44,45] has only 3 constituents—salvianolic acids, ginsenosides, and ginseng polysaccharides. Consequently, it is not surprising that the pharmacological profile of the combination RG–RS is unique and different from other substances observed in other studies. It is noteworthy that, in our study, we have observed an effect on the transcriptomic level of regulation of cellular response in vitro model; our experiments did not cover many other effects on the proteomic level of regulation, including effects on the enzymatic activity of proteins, interactions with receptors, and other functions of proteins.

The results of this study, for the first time, describe the effect of RG–RS on the gene expression profile of isolated neurons, the potential impacts of RG–RS on diseases and disorders, biological processes, cellular and physiological function, and implications on intracellular signaling and metabolic pathways (Figure 5).

In this study, we, for the first time, demonstrate the inhibitory effect of RG–RS on gene expression regulating glycolysis metabolic enzymes (Figure 6 and Appendix A), which is crucial in cancer.

Most cancer cells are characterized by a high rate of aerobic glycolysis for the energy supply of proliferation, tumor growth, and survival, even with sufficient oxygen supply [59,60,61,62]. Highly proliferating cancer cells undergo the Warburg effect, i.e., a metabolic switch from oxidative phosphorylation providing 36 molecules of energy molecules (ATP) to an abnormal and ineffective energy supply metabolic pathway, aerobic glycolysis, providing only 4 ATP molecules and mostly lactate instead of pyruvate (Figure 6). The glucose metabolic pathway is activated by several tumor-promoting (P3IK/AKT, HIF-1α, mTOR, K-Ras, c-Myc, miRNAs,) and inhibited by tumor-suppressing (AMPK, p53) signaling pathways in highly proliferative cancer cells [60,62].

Several transcription factors and signaling molecules, such as HIF-1α, c-Myc, Akt, and mTOR, and other upstream regulators, including oncogene K-Ras, tumor suppressor p53, energy sensor adenosine monophosphate-activated protein kinase (AMPK), non-coding RNAs, and sirtuin family proteins and deacetylation, are also involved in this process [59,60,61,62]. Decreased glycolytic flux in response to AMPK or p53 may be an adaptive response to shut off proliferative metabolism during periods of oxidative stress or low energy availability [59].

Figure 6 and Appendix A show that RG–RS strongly downregulates the expression of five genes upregulated in cancer cells: 2562-fold downregulated neuron-specific enolase (NSE) ENO1 gene, 1426-fold downregulated phosphoglycerate kinase 1 PGK1 gene, 1094-fold downregulated pyruvate kinase PKM gene, 855-fold downregulated aldolase ALDOA gene, and 810-fold downregulated triose phosphate isomerase TPI gene. Moreover, RG–RS significantly inhibits the HIF-1α signaling pathway, a master regulator of glycolysis, and activates antitumor monophosphate-activated protein kinase (AMPK), suggesting the substantial inhibition of the conversion of glucose into lactate (Warburg effect) in cancer, which is crucial for the proliferation and survival of tumor cells.

Figure 6 shows that RG–RS targets 10 (!) disease-specific molecules (5 directly and 5 indirectly via upstream regulators HIF-1α and AMPK) in the tumor’s energy supply source.

Overall, RG–RS targets 1028 targets associated with cancer and 139 with metastasis, Table 1 and Appendix A. In silico IPA analysis of the effects of RG–RS on gene expression in isolated neurons predicts the antitumor activity of RG–RS, which is in line with other studies [42,43,44,45].

Figure 2, Figure 3 and Figure 4 show the potential effects of RG–RS on the canonical signaling pathways, cellular and physiological functions, and aging-related diseases, particularly the inhibition of the osteoarthritis signaling pathway where the probability of prediction is significant (−log *p*-value 1.7, z-score -3.2; Appendix A). That is in line with other studies of RG and RS where the potential health effect of Reg Ginseng [63,64,65,66,67] and Red Sage [68,69,70,71,72] in osteoarthritis was demonstrated.

A limitation in this study is related to the lack of scientific literature about the direction (negative or positive) of correlations between gene expression and physiological function or disease for predicting the effects of some experimental data used in silico analysis.

## 4. Materials and Methods

All the materials and methods used in this study have been recently described in detail in our recent publication [24], which was conducted by the same experimental protocol [24]. They include mRNA microarray hybridization conducted by the same experimental protocol, Ingenuity Pathway Analysis (IPA), and Statistical Analysis [24].

### 4.1. Test Samples and Their Concentrations in Murine Hippocampal Neuronal HT22 Cells Culture

*Panax ginseng* C.A. Mey. (Korean Ginseng), radix and *Salvia miltiorrhiza* Bunge (Danshen/Red Sage), radix et rhizome were hydroponically cultivated in controlled conditions at Botalys S. A. (Ath, Belgium). Korean Ginseng was steamed to Red Ginseng, which was dried, powdered, and standardized for the content of ginsenosides Rk1 and Rg5 to obtain Red Ginseng HRG80TM preparation containing 7.6% of total ginsenosides. HRG80TM preparation was exhaustively extracted by 70% methanol and evaporated to dryness to obtain HRG extract (DER 4: 1) containing 1.9% Rg5 and 1.0% Rk1; the content of total ginsenosides in the HRG extract was 30.32%.

Red Sage radix et rhizomatic were extracted by 70% methanol to obtain salvianolic acid-free extract (RS) containing 0.883% cryptotanshinones, 0.673% tetrahydrotanshinone, 0.094% dihydrotanshinones, and 0.029% tanshinone IIA; total tanshinons content—1.72%.

Murine hippocampal neuronal HT22 cells were incubated with test samples at final concentrations of 10 μg/mL of RG and 10 μg/mL RS in RG–RS, 10 μg/mL of RG, 0.1 µM of ginsenosides Rb1, Rg3, Rg5, and Rk1 (Table 1) or DMSO as solvent control (0.5%) for 24 h.

Purified reference standards of ginsenosides were purchased from Merck KGaA (Darmstadt, Germany) and Extrasynthese (Genay, France).

HPLC conditions. Analytical samples were analyzed by Waters LC system using reverse phase HPLC column ACE C18 5 μm (250 × 4.6 mm) and the solvent system containing gradually increasing concentration (from 21 to 90% in eight steps) of acetonitrile in water solution of 0.1% ortho-phosphoric acid at 30 °C, flow rate-1.2 mL/min.; diode array detection at 203 nm. The content of analytical markers in the analytes was quantified by reference standards used to generate calibration curves. The analytical method was validated for selectivity, precision, and accuracy (RSD < 5%).

### 4.2. mRNA Microarray Hybridization

Murine hippocampal neuronal HT22 cells were seeded and attached for 24 h before drug treatment. Cells were then treated for 24 h at final concentrations of 10 μg/mL of RG and 10 μg/mL RS in RG–RS, 10 μg/mL of RG, 0.1 µM of ginsenosides Rb1, Rg3, Rg5, and Rk1 (Table 1) or DMSO as solvent control (0.5%). The total RNA Nanochip assay was applied on an Agilent 2100 bioanalyzer (Agilent Technologies GmbH) for quality control of RNA. An RNA index value > 8.5 served as a threshold for further processing of RNA samples. Every two independent experiments have been performed. The mRNA microarray hybridization has been performed at the Genomics and Proteomics Core Facility, German Cancer Research Center (Heidelberg, Germany), using Affymetrix GeneChips^®^ with mouse Clariom S assays. A quantile normalization algorithm has normalized the measurements without background subtraction. The standard deviation differences were calculated in one-by-one comparisons to identify differentially regulated genes. The Chipster software (The Finnish IT Center for Science CSC) was used for further evaluation of the results.

### 4.3. Ingenuity Pathway Analysis (IPA)

The interpretation of microarray data and gene expression changes was performed using Ingenuity Pathways Analysis (IPA) software, summer release 2021 (QIAGEN Bio-informatics, Aarhus C, Denmark), which relies on the Ingenuity Knowledge Base, a continuously updated database gathering observations with more than 8.1 million findings manually curated from the biomedical literature or integrated from 45 third-party databases. The IPA network contains 40,000 nodes representing mammalian genes, molecules, and biological functions, linked by 1,480,000 edges representing experimentally observed cause–effect relationships (either activating or inhibiting) related to gene expression, transcription, activation, molecular metabolism, and binding. Network edges are also associated with a direction (either activating or inhibiting) of the causal effect [73].

Using the IPA Core Analysis tool for all tested transcriptomic datasets, we performed predictive analyses of the impact of test samples on canonical signaling and metabolic pathways, which displayed the molecules of interest within well-established pathways; and diseases, disorders, molecular and cellular functions that are activated or inhibited downstream and upstream of the genes, whose expression has been altered.

### 4.4. Statistical Analysis

Two statistical methods of analysis of gene expression datasets were used in the IPA: (i) the gene-set-enrichment method, where differentially expressed genes are intersected with sets of genes that are associated with a particular biological function or pathway providing an “enrichment” score (Fisher’s exact test *p*-value) that measures the overlap of the observed and predicted regulated gene sets [74,75], (ii) the method based on cause–effect relationships related to the direction of effects reported in the literature [76,77] that provides the so-called z-score measuring the match of observed and predicted up/down-regulation [73]. The predicted effects are based on gene expression changes in the experimental samples relative to the control; z-score > 2, −log *p*-value > 1.3.

## 5. Conclusions

The results of this study present evidence of the potential efficacy of the fixed combination of Red Ginseng (*Panax ginseng* C.A. Mey.) and Red Sage/*Danshen* (*Salvia miltiorrhiza* Bunge) in cancer. Further preclinical and clinical studies are warranted to assess the efficacy and safety of RG–RS in preventing the progression of cancer and other aging-related diseases.

## Figures and Tables

**Figure 1 pharmaceuticals-15-01345-f001:**
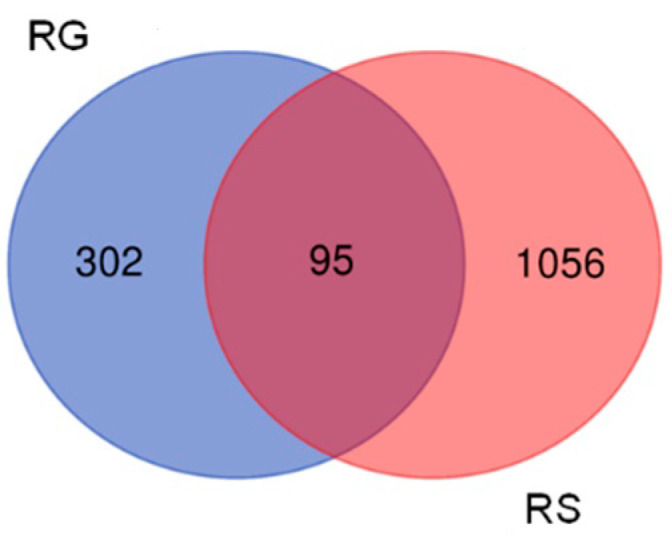
Venn diagrams the number of genes deregulated by RG and RG–RS in the hippocampal neuronal cell line HT22.

**Figure 2 pharmaceuticals-15-01345-f002:**
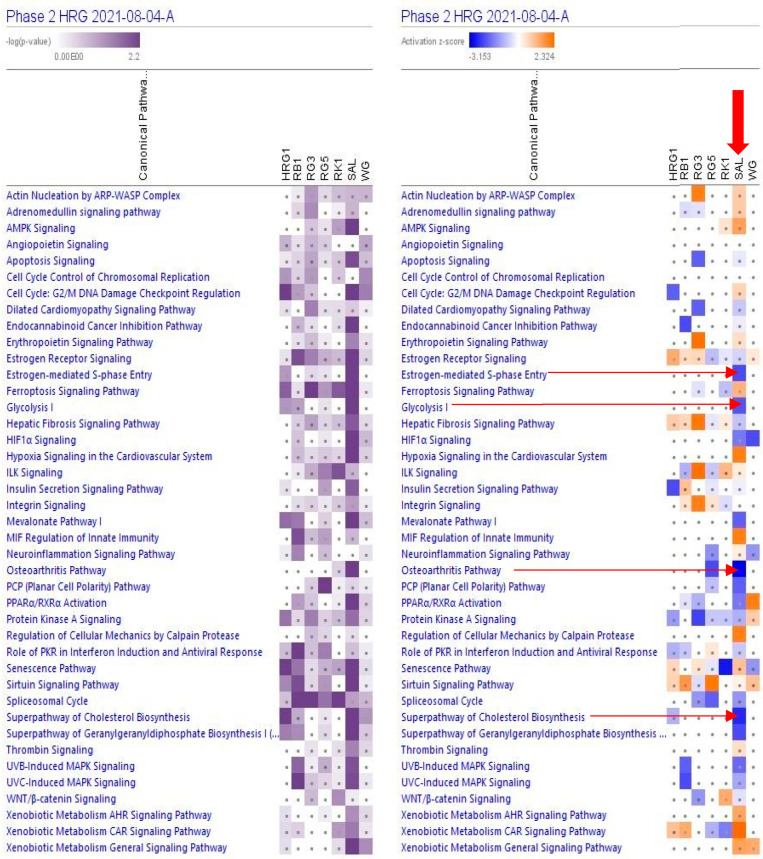
Effects of RG–RS (SAL) at concentrations of 20 μg/mL, RG at concentrations of 10 μg/mL (HRG1), white ginseng (WG), and ginsenosides Rb1, Rg3, Rg5, Rk1 at the concentration of 0.1 μM on selected canonical pathways. The violet color shows the *p*-value (calculated using a Right-Tailed Fisher’s Exact Test), indicating the statistical significance of the overlap of analyzed dataset genes within the pathway; the intensity of color corresponds to the log *p*-value in a scale from 0 to 2.2 (upper panel). The brown color shows the predicted activation, and the blue color shows the predicted inhibition of signaling pathways; the intensity of the color corresponds to the activation zone on a scale from −3.153 to +2.324 (upper panel). symbol · shows that the activation z-score was < 2 and the −log *p*-value <1.3 = *p* < 0.05. An absolute z-score ≥2 is considered significant activation (+) or inhibition (-). The activation z-score predicts the activation state of the canonical pathway using the gene expression patterns of the genes within the pathway.

**Figure 3 pharmaceuticals-15-01345-f003:**
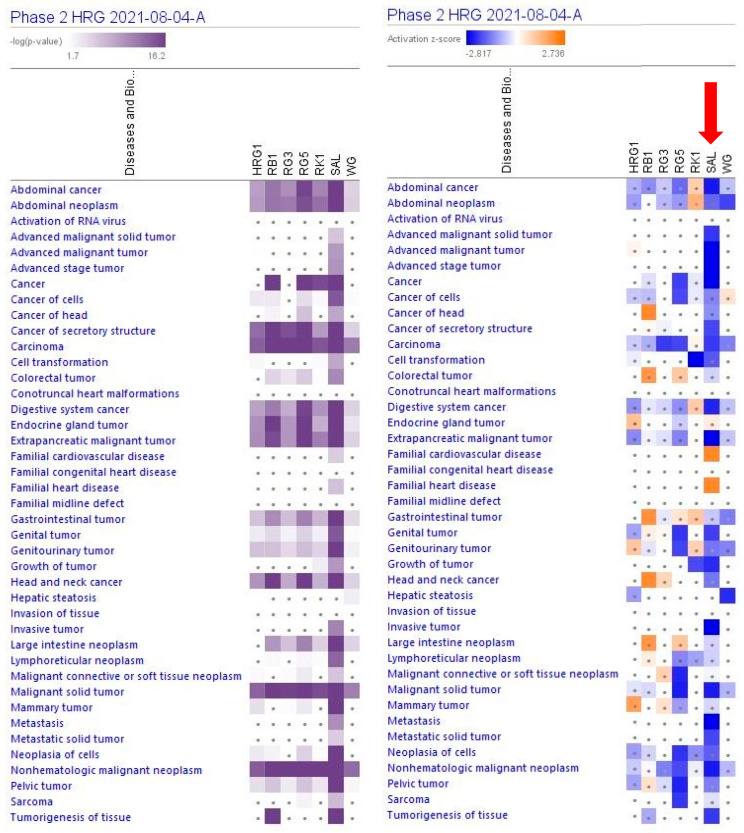
Predicted effects of RG–RS (SAL) at concentrations of 20 μg/mL, RG at concentrations of 10 μg/mL (HRG1), white ginseng (WG), and ginsenosides Rb1, Rg3, Rg5, Rk1 at a concentration of 0.1 μM. The violet color shows the *p*-value (calculated using a Right-Tailed Fisher’s Exact Test), indicating the statistical significance of the overlap of analyzed dataset genes within the diseases; the intensity of color corresponds to the log *p*-value in a scale from 1.7 to 16.2 (upper panel). Disease scores are shown using a gradient from dark blue to brown for predicted activation and light to dark blue for predicted inhibition of diseases on a scale from −2.817 to +2.736 (upper panel); symbol · shows that the activation z-score was < 2, and the −log *p*-value <1.3 = *p* < 0.05. An absolute z-score ≥ 2 is considered significant activation (+) or inhibition (-).

**Figure 4 pharmaceuticals-15-01345-f004:**
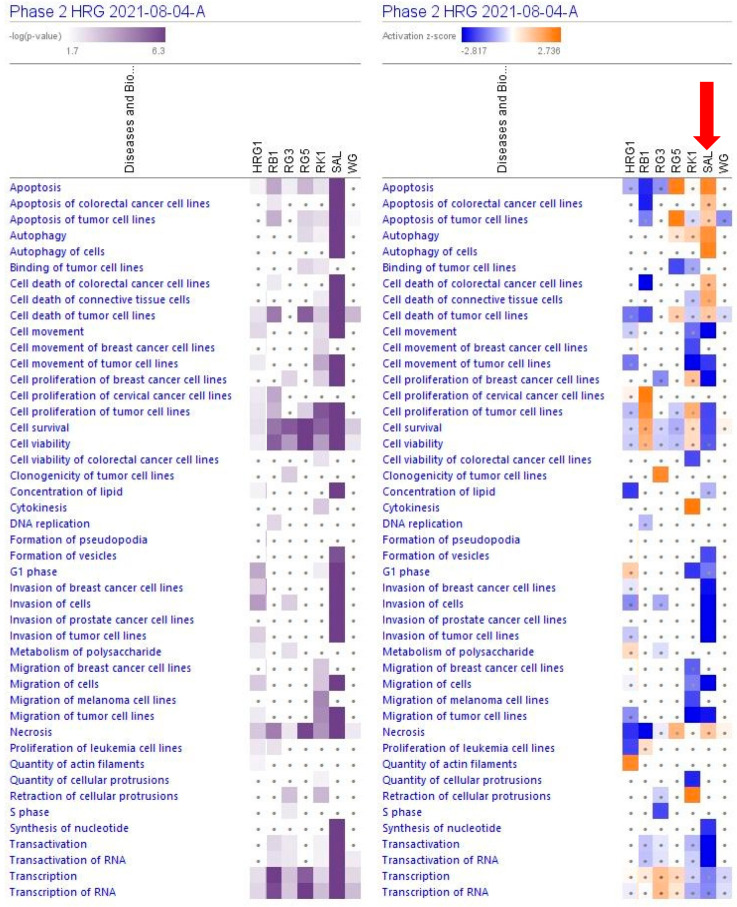
Predicted effects of RG–RS (SAL) at a concentration of 20 μg/mL, RG at a concentration of 10 μg/mL (HRG1), white ginseng (WG), and ginsenosides Rb1, Rg3, Rg5, Rk1 at a concentration of 0.1 μM on cellular and physiological functions. The violet color shows the *p*-value (calculated using a Right-Tailed Fisher’s Exact Test), indicating the statistical significance of the overlap of analyzed dataset genes within the functions; the intensity of color corresponds to the log *p*-value in a scale from 1.7 to 6.3 (upper panel). The function scores are shown using a gradient from dark blue to brown for predicted activation and light to dark blue for predicted inhibition of diseases; the intensity of color corresponds to the activation zone on a scale from −2.817 to +2.736 (upper panel). symbol · shows that the activation z-score was < 2 and the −log *p*-value <1.3 = *p* < 0.05. An absolute z-score ≥2 is considered significant activation (+) or inhibition (-). The activation z-score predicts the activation state of the function using the gene expression patterns.

**Figure 5 pharmaceuticals-15-01345-f005:**
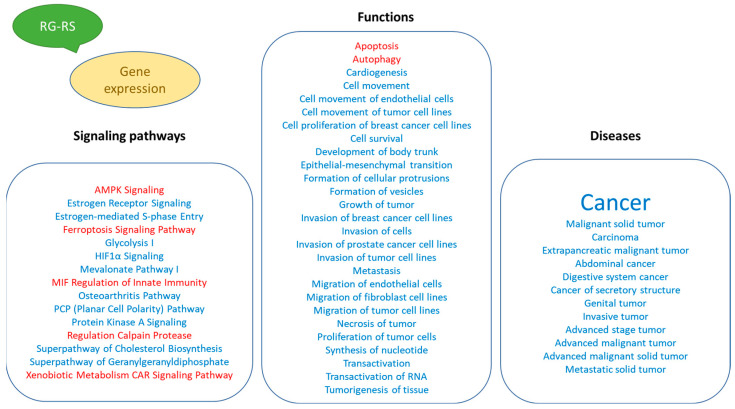
Potential effects of RG–RS on diseases and disorders, biological processes, cellular and physiological function, and detected impact on intracellular signaling and metabolic pathways.

**Figure 6 pharmaceuticals-15-01345-f006:**
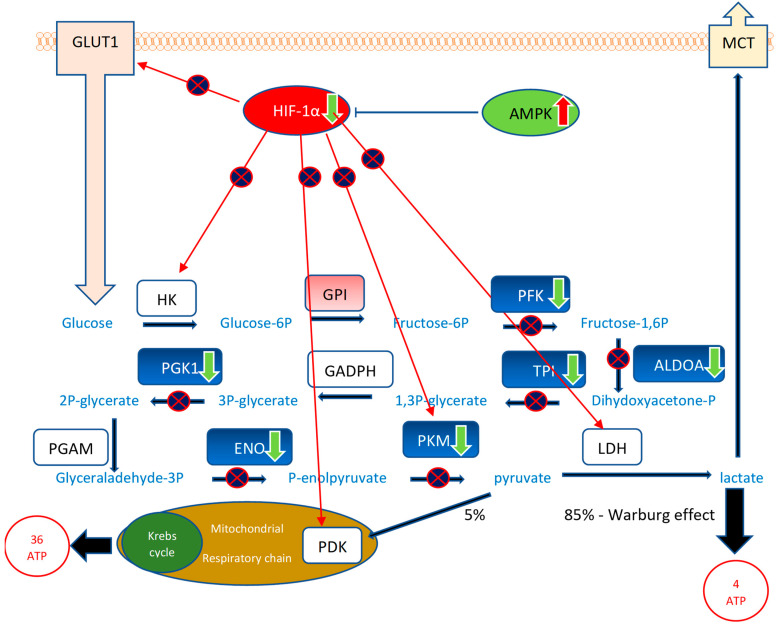
The effect of RG–RS on glycolysis: RG–RS strongly downregulates the expression of 8 genes encoding eight enzymes of the glucose metabolic pathway, significantly inhibits the HIF-1α signaling pathway, a master regulator of glycolysis, and activates antitumor monophosphate-activated protein kinase (AMPK) suggesting substantial inhibition of the conversion of glucose into lactate (Warburg effect) in cancer that is crucial for proliferation and survival of tumor cells. Downregulated genes are shown in the green color, corresponding enzymes in the blue color: the neuron-specific enolase (NSE) ENO1 gene (−2562-fold change), phosphoglycerate kinase 1 PGK1 gene (−1426-fold change), the pyruvate kinase PKM gene (−1094-fold change), the aldolase ALDOA gene (−855-fold change), the triose phosphate isomerase TPI gene (−810-fold change), phosphofructokinase, encoded by the PFK gene (−730-fold change). HKs—hexokinases HK1, HK2, HK3, HK4, which are encoded by the HK1, HK2, HK3, and HK4 genes; GPI—glucose-6-phosphate isomerase encoded by the GPI gene (+372-fold change); GAPDH: glyceraldehyde 3-phosphate dehydrogenase, encoded by the GAPDH gene; PGAM—phosphoglycerate mutase, encoded by the PGAM gene; LDH—lactate dehydrogenases; which are encoded by LDHA and LDHB genes; GLUT1—glucose transporting proteins encoded by the SLC2 gene; MCTs: lactate transporters encoded by the SLC16 gene. Tumor suppressors are shown in green, and oncogenes are in red. RG–RS inhibits (*p* > 1.3; z 1.5, Appendix A) the oncogenic Hypoxia-inducible factor-1 (HIF-1α) and activates (*p* > 1.3; z—1.7, Appendix A) tumor suppressor adenosine monophosphate-activated protein kinase (AMPK) signaling pathways.

**Table 1 pharmaceuticals-15-01345-t001:** Final concentrations of RG–SG, RG, and their active constituents, ginsenosides Rb1, Rg3, Rg5, and Rk1 in the hippocampal neuronal cell line HT22, and of the number of deregulated genes (>20 fold compared to control).

Sample Name	Concentration μg/mL	Content in RG–RS Dry Extract, %	Number of Deregulated Genes	Substance Specific Genes *
RG–RS	20 (10/10)	100	1151	1056
RG	10	50	397	397
Rb1	0.0111–0.1 μM	0.09	470	279
Rg3	0.0785–0.1 μM	2.15	413	236
Rg5	0.0767–0.1 μM	3.77	553	345
Rk1	0.0767–0.1 μM	2.0	373	214
TT SA	0.172–0.58 μM ** 0	0.86 0	n/a n/a	n/a n/a

TT—total tanshinones; SA—salvianolic acid B; *—gene numbers excluding those deregulated by RG; **—calculated as tanshinone II (MW 294.3).

## Data Availability

Data is contained within the article and Appendix A.

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
