# Peer review of "Network Pharmacology of Ginseng (Part III): Antitumor Potential of a Fixed Combination of Red Ginseng and Red Sage as Determined by Transcriptomics"

_pharmaceuticals, 2022, doi:10.3390/ph15111345_

Round 1

Reviewer 1 Report

The authors demonstrated gene expression profile of fixed combination of Red Ginseng and Red Sage belonging to Traditional Chinese Medicine within murine HT22 hippocampal cell culture. Further gene expression fingerprinting was proceeded through network pharmacology Ingenuity pathway analysis highlighting their mechanistic aspects as anti-cancer agent. Findings are interesting with significant scientific soundness. Publication of this manuscript is recommended following minor revision;

1.      Introduction section, lines 43-51, correlation between neuroinflammation and cognitive deficits could be more elaborated within the literature review highlighting the traditional application of Red Ginseng with obtained in vivo and in vitro neuroprotective properties.

2.      The rational for adopting the particular neuronal cell (murine HT22 hippocampal cells) rather than other type of cells for assessing gene expressions, should be thoroughly discussed by the authors.

3.      Table 1. The last column is quite confusing and needs to be more clarified. It could be better to state that the column represents the number of deregulated genes that were not deregulated by RG (i.e. * gene numbers excluding those deregulated by RG).

4.      Also, in Table 1, it is recommended to remove data for “salvianolic acid B” from the Table since it was not tested.

5.      In Figures 2-4, the authors only indexed colors for Z-scores at the figure legends, while -Log P-values were left uncharted. Authors are advised to comment on Lop P-values as well. Additionally, the figure color coding needs more description since they are graded color scales (dark / faint, blue and brown colors). So, graded color bar scale is best to describe these figures.

6.      Resolution of Figure 4 should be improved. Text is obscured and tiny to be recognized.

7.      Authors should elaborate more on the discussion part comparing the obtained results (gene expression profiles) with reported experimental data or similar studies investigating the same or close class analogues.

8.      Within the discussion part, authors elaborated on the role of target genes in cancer signaling, while missing their mechanistic role within age-related diseases non-described, despite being the main traditional use of the investigated natural plant in traditional medicine.

9.      Typo need correction, as in “0,75 g” line 59, and abbreviations like CRP and FMG, lines 56 and 67, should be initially annotated before being inserted within the context. 

Reviewer 2 Report

In the reviewer’s opinion, the authors have long-term, extensive experience in the discussed subject and in research methodology. The obtained results, and in particular the discussion of obtained results and possible causes, were very interesting.

The reviewer does not have any other comments, they would be a bit more acute and would not contribute to the enormous amount of information and contribution made by the team to the achievement of the presented results.

Below, I’ve presented only few very minor editorial comments, suggestions lets say, which, in the opinion of the reviewer, will facilitate the reading of the text of the manuscript for the reader.

Particularly noteworthy are the very well conducted experimental work, and a very interesting and extensively described discussion. thus, such brief and matter-of-fact discussion, along with a critical look at the further potential of the investigation, deserves praise.

Other comments, in the opinion of the reviewer, would not bring much scientific attention, but could rather be of a different nature than those mentioned.

4. Materials and Methods

4.1. Test Samples and their concentrations in murine hippocampal neuronal HT22 cells culture

1. In the opinion of the reviewer, despite the mention of the valuable results of the previous publications of the team, a short discussion of the source of the tested preparation, the reference preparation, and the method of obtaining and standardizing them - at the end of the sentence, a relevant publication source should be cited here indicating the exact methodology of extraction, preparation, standardization and phytochemical analysis.

2. According to the reviewer, there is no shortlisting of the individual stages of molecular analysis - gene expression using the microarray hybridization technique: was the procedure performed using commercial kits? on which manufacturer's system was the hybridization analysis performed? The same applies to the remark regarding the methodology for the stages: Ingenuity Pathway Analysis (IPA) and Statistical Analysis. 

3. I propose to, briefly, indicate the purpose of using this particular cell line presented in the manuscript. What would this cell line represent as a model?

Please consider these comments. Although they are not, in a way, essential (as the authors refer to previous publications and the team's experience) - this type of information, albeit brief, at this stage of the manuscript - will be of great help to any reader of the manuscript in understanding the transparency and reliability of the preparations and analytical methodology. 

Reviewer 3 Report

1.Why do you choose the murine HT22 hippocampal cell pointedly to built the experimental model? Please list the original reference.

2.In TCM theory, the effects of Red Ginseng and Danshen are produced through the brain vessel, but the paper has not performed the reletive research. In vivo experiments were missing, too.

3.You use the experimental thersold of RG-SG and RG 20 fold and 3 fold respetively. Wouldyou like to present the statistical evidence? Otherwise, the effects of Rb1, Rg3, Rg5, Rk1 seens to be not considered independently?

4. And the concerntrantion on the two coponent, how to determine that?

5.The conclusion described a crudly theory that RG-SG and RG may improve many diseases' treatment throgh lots of target. I think it shoule be more considerable. It is not clear nor accurate enough.

6. Figure legends were not clear enough to explain them. Please describe them more clear.

7. Some figures are blur, please upload clear ones.

Round 2

Reviewer 3 Report

I suggest this manuscript can be published after the minor language revision.